# Protein Language Model-Powered 3D Ligand Binding Site Prediction from Protein Sequence

**Shuo Zhang**[1,3]**, Lei Xie**[1,2,3]
[1]Department of Computer Science, Hunter College, City University of New York
[2]Ph.D. Program in Computer Science, Graduate Center, City University of New York
[3]Helen & Robert Appel Alzheimer's Disease Research Institute,
Feil Family Brain & Mind Research Institute, Weill Cornell Medicine, Cornell University
szhang4@gradcenter.cuny.edu, lei.xie@hunter.cuny.edu

## Abstract

Prediction of ligand binding sites of proteins is a fundamental and important task for understanding the function of proteins and screening potential drugs. Most existing methods require experimentally determined protein holo-structures as input. However, such structures can be unavailable on novel or less-studied proteins. To tackle this limitation, we propose LaMPSite, which only takes protein sequences and ligand molecular graphs as input for ligand binding site predictions. The protein sequences are used to retrieve residue-level embeddings and contact maps from the pre-trained ESM-2 protein language model. The ligand molecular graphs are fed into a graph neural network to compute atom-level embeddings. Then we compute and update the protein-ligand interaction embedding based on the protein residue-level embeddings and ligand atom-level embeddings, and the geometric constraints in the inferred protein contact map and ligand distance map. A final pooling on protein-ligand interaction embedding would indicate which residues belong to the binding sites. Without any 3D coordinate information of proteins, our proposed model achieves competitive performance compared to baseline methods that require 3D protein structures when predicting binding sites. Given that less than 50% of proteins have reliable structure information in the current stage, LaMPSite will provide new opportunities for drug discovery.

## 1 Introduction

Identifying ligand binding sites of proteins is an essential step in the pipeline of drug discovery [1]. The binding site serves as a key interface where proteins engage in fundamental cellular processes, making it an attractive target for drug molecules. To reduce the time and expense of binding site identification for protein-ligand complexes, computational methods have been proposed and achieved promising performance [2, 3, 4]. Non-machine learning computational methods include geometry-based ones [5, 6, 7, 8, 9, 10, 11, 12] that identify and rank hollow spaces on protein surfaces, probe-based ones [13, 14, 15, 16] that strategically place probes on protein surfaces to compute energies for identifying binding locations, and template-based ones [17, 18, 19] that query protein templates in large databases with annotated binding sites. Machine learning methods have further advanced the field, leveraging various techniques, including classic machine learning algorithms [20, 21] and deep learning models [4]. Notably, 3D-convolutional neural networks have been widely used by treating the task of binding site identification as a segmentation problem within 3D space [22, 23, 24, 25, 26, 27, 28].

Despite the progress in ligand binding site prediction, existing methods predominantly depend on the availability of experimentally determined ligand-bound (holo) protein structures, which pose

certain limitations. Firstly, these methods encounter challenges when dealing with newly sequenced proteins lacking experimental structural data, hindering their applicability to a broader range of protein targets. Secondly, for proteins with only ligand-free (apo) structures available, detecting cryptic binding sites—those whose shapes may change upon ligand binding—can be a formidable task [29]. Recently, there have been breakthroughs in predicting highly accurate protein structures, which provide extra sources of structural information besides experimental techniques [30, 31, 32, 33]. A growing number of works have applied binding site prediction methods on these computationally achieved protein structures [34, 35, 36]. However, due to the limited portion of predicted protein structures with high confidence and the ignorance of the dynamic nature of binding sites in predicted proteins, experimentally determined structures still provide better and easier situations for binding site identification and docking [37].

To address the aforementioned limitations, we have a two-fold objective. Firstly, we seek to leverage the computationally predicted protein information to expand our capability to predict binding sites on a broader set of proteins, including those lacking experimental structural data. Secondly, we aim to reduce the heavy dependence on rigid 3D protein structures and instead incorporate the dynamic nature of binding sites. To achieve these goals, we propose Language Model-Powered Ligand Binding Site Predictor (LaMPSite), which capitalizes on the protein information computed from protein sequence based on the recent ESM-2 pre-trained protein language model [33]. The computed protein information includes residue-level embeddings that implicitly capture the 3D structure of a protein. These embeddings are integrated with atom-level embeddings of ligands obtained from a graph neural network to compute the protein-ligand interaction embedding. We also utilize the predicted protein contact map from ESM-2 and the distance map of ligand conformer as geometric constraints to further update the protein-ligand interaction embedding to learn more realistic binding site structures. We then apply mean pooling to the interaction embedding, enabling us to quantify the score between each protein residue and the entire ligand, where a higher score indicates a greater likelihood that the residue is part of a binding site. Finally, we leverage the protein contact map information to guide the clustering of filtered residues, ultimately yielding our residue-level predictions for binding sites.

We evaluate our LaMPSite against several baselines for binding site prediction on a benchmark dataset. Notably, our method demonstrates competitive performance without the need for 3D protein coordinate information, in contrast to baseline methods that heavily rely on 3D protein structures. The results highlight that LaMPSite provides a novel and promising direction for binding site prediction, driven by the capabilities of protein language models. Our main contributions are as follows:

- We introduce LaMPSite, a novel ligand binding site prediction method powered by a protein language model. LaMPSite relies solely on protein sequences and ligand molecular graphs as initial input and eliminates the need for 3D protein coordinates throughout the prediction process.

- When compared to baseline methods that utilize experimental 3D protein holo-structures, our model demonstrates competitive performance on a benchmark dataset for binding site prediction.

## 2 Method

The overall architecture of LaMPSite is depicted in Figure 1. We will describe the relevant modules in this section. Additional details that are not covered here are included in the Appendix.

### 2.1 Involved Representations

**Protein.** Our model only takes protein sequences as the initial input to compute protein representations. Leveraging a pre-trained ESM-2 protein language model, we generate protein residue embeddings $\boldsymbol{h}^p \in \mathbb{R}^{n_p \times d}$ from the provided protein sequence, where $n_p$ is the number of residues in the sequence and $d$ is the hidden dimension size. These embeddings are directly derived from ESM-2 and have demonstrated the emergence of 3D structural information inside [33]. Additionally, we make use of the unsupervised contact prediction results obtained from ESM-2 to generate protein contact maps $\boldsymbol{C}^p \in \mathbb{R}^{n_p \times n_p}$, serving as a low-resolution estimate of the protein structural information.

**Ligand.** The 2D ligand molecular graphs excluding hydrogen atoms are initially provided to be fed into a Graph Neural Network [38] to learn atom-level embeddings $\boldsymbol{h}^l \in \mathbb{R}^{n_l \times d}$, where $n_l$ is the

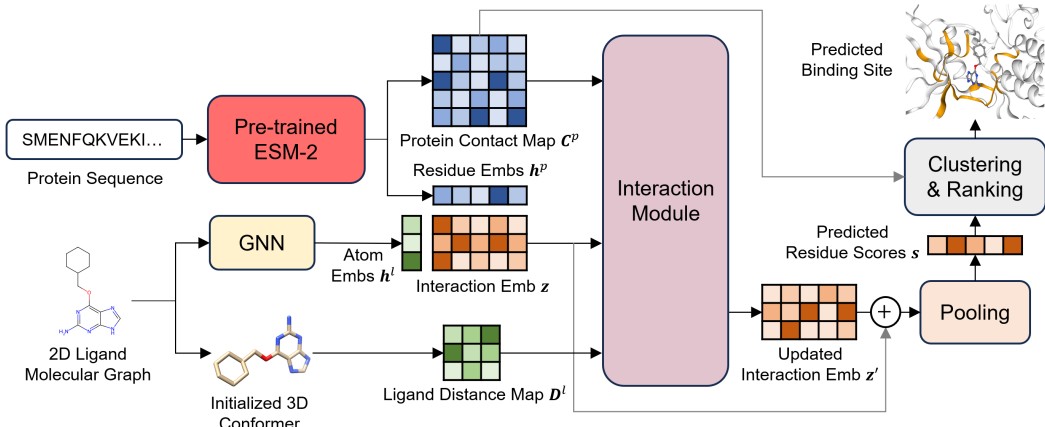

Figure 1: **Illustration of LaMPSite pipeline.** The overall pipeline involves taking a protein sequence and a 2D ligand molecular graph as input. Initially, the pre-trained ESM-2 model computes protein residue embeddings $h^p$ and a contact map $C^p$. Subsequently, ligand atom embeddings $h^l$ are acquired through a GNN and are then combined with residue embeddings to calculate the interaction embedding $z$ for the protein-ligand pair. $z$ is refined in the interaction module using geometric constraints, including $C^p$ and the ligand distance map $D^l$ from the initialized 3D conformer. Then the interaction embeddings are aggregated, followed by pooling to generate scores $s$ for each residue. Finally, residues are clustered based on the protein contact map, and these clusters are ranked to determine the binding site.

number of ligand atoms. Besides, we use RDKit [39] to initialize a 3D conformer for each ligand to compute the corresponding ligand distance map $D^l \in \mathbb{R}^{n_l \times n_l}$.

**Protein-ligand pair.** Based on the protein residue embeddings $h^p$ and ligand atom embeddings $h^l$, we compute the interaction embedding $z \in \mathbb{R}^{n_p \times n_l \times d}$ for the protein-ligand pair, which models the interactions between protein residues and ligand atoms. For $i$-th protein residue and $j$-th ligand atom, we define $z_{ij} = h_i^p \odot h_j^l$. This information inherently reflects the residues with stronger interactions with ligand atoms, thereby identifying residues within binding sites.

## 2.2 Interaction Module

While the aforementioned $z$ can be directly employed to predict binding site residues, it's worth noting that the 3D structural information that emerged from ESM-2 may not be ideal holo-structures for detecting binding sites. Therefore, we want to incorporate the dynamic nature of binding sites into our model, which involves introducing further modifications to $z$. To accomplish this, we use the trigonometry module in [40], which is a variant of the Evoformer block in [30], as our interaction module to update $z$ based on the geometric constraints in protein contact map $C^p$ and ligand distance map $D^l$.

## 2.3 Pooling Module

After having the updated interaction embedding $z'$, we sum it together with the original $z$ and perform a linear transformation to reduce the hidden dimension size to 1. The resulting final interaction embedding $z^{final} \in \mathbb{R}^{n_p \times n_l \times 1}$ contains scores that indicate the likelihood of interaction between residues and ligand atoms. To compute a score $s$ for each residue, representing the interaction likelihood between the entire ligand and the residue, we apply mean pooling specifically over the dimension related to ligand atoms on $z^{final}$. For the $i$-th residue, $s_i = \sum_j z_{ij}^{final}$, where $j$ represents the $j$-th ligand atom. A higher score indicates a stronger interaction between a residue and a ligand, concurrently signifying that the residue likely constitutes part of the binding site.

### 2.4 Clustering and Ranking Module

For the identification of binding sites on the test set, we first normalize the predicted scores $s$ to the range of 0 to 1. Following this, we apply a threshold $v$ to filter out residues with scores less than $v$. Subsequently, we cluster the remaining residues using the single linkage algorithm [41], utilizing the protein contact map generated by ESM-2. The clusters are finally ranked based on the squared sum of the associated residue scores. An illustration of these steps can be found in Appendix A.1.

## 3 Experiments

**Datasets.** For the training of our model, we use the scPDB v.2017 database [42], which is a widely used dataset for binding site identification. It consists of 17594 binding sites, corresponding to 16612 PDB structures and 5540 UniProt IDs. To ensure a rigorous evaluation process, we adhere to the same data split strategy employed in [24, 26], which prevents any data leakage, and utilizes a 10-fold cross-validation. Multi-chain 3D complexes are split into single chains with their own interacting ligands, and each split pair is considered as a protein-ligand pair. To reduce memory usage, we remove protein sequences longer than 850 amino acids. The ground-truth label of each residue is defined as whether the distance between the residue and the nearest ligand atom is $< 8$ Å.

For the evaluation of our model, we use COACH420 [21], which is a data set with 420 proteins that contain a mix of drug targets and natural ligands. Specifically, we use the version referenced in [26], which excludes ligands that were improperly prepared or failed to be parsed, resulting in 291 protein structures and 359 ligands. To prevent any potential data leakage, we follow [26] to exclude structures in scPDB that have either sequence identity $> 50\%$ or ligand similarity $> 0.9$ and sequence identity $> 30\%$ to any of the structures in COACH420. Consequently, our processed scPDB dataset contains 16270 protein-ligand pairs. The data sources are provided in Appendix A.2.

**Experimental Settings.** For the training process, our model is optimized to minimize the binary cross-entropy loss between the predicted residue scores $s$ and the ground-truth labels. An early-stopping strategy is adopted to decide the best epoch based on the validation loss.

For our evaluation criteria, we use the DCA criterion, which stands for the distance from the center of the predicted binding site to the closest ligand heavy atom. DCA criterion evaluates the model's ability to find the location of the binding site. Given that our model predicts the residues belonging to binding sites, we use the ground-truth protein structure to calculate the predicted binding site center by averaging the coordinates of all alpha-carbons in our predicted binding site residues. Predictions with DCAs $< 4$Å are considered successful. Specifically, we evaluate the model's ranking capability by measuring the success rates when considering the top n ranked pockets, where n is the number of annotated binding pockets for a given protein. We calculate the success rates for each fold in the 10-fold cross-validation, and the final result is obtained by averaging these success rates.

We compare our model with the following baselines: Fpocket [10], Deepsite [22], Kalasanty [24], DeepPocket [26], P2Rank [21]. The results of the baselines are from [26], which are all evaluated on the same test set as ours. Among the baselines, Fpocket is a geometry-based method, Deepsite, Kalasanty, and DeepPocket are 3D-CNN-based methods, and P2Rank is a random forest-based method. All baselines require the experimental protein holo-structures as input. More details of the implementations can be found in Appendix A.3.

## 4 Results

To evaluate the binding site identification performance of LaMPSite, we first plot the relationship between DCA success rates and distance thresholds in Figure 2(a). LaMPSite returns an average DCA success rate of 66.02% at 4Å threshold. Comparing this result with baseline methods, we observe that LaMPSite significantly outperforms Fpocket, Deepsite, and Kalasanty, while trailing DeepPocket (67.96%) and P2Rank (68.24%), as illustrated in Figure 2(b). These results demonstrate that LaMPSite, which does not rely on 3D protein coordinate information for binding site prediction, effectively leverages the 3D structural information that emerged in the pre-trained ESM-2 model. Furthermore, it's noteworthy that LaMPSite consistently provides at least one binding site prediction for each protein in COACH420, a behavior similar to Fpocket, DeepPocket, and P2Rank. In contrast, Deepsite fails for one protein, and Kalasanty fails for 12 proteins. This indicates the

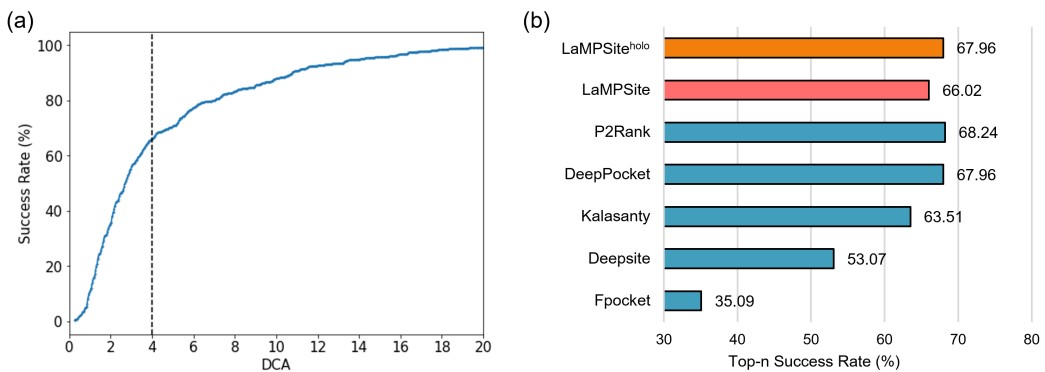

Figure 2: **Models' performance on COACH420.** (a) Success rate plot for different DCA thresholds for LaMPSite. (b) Comparison of identification performance (Top-n success rate) in terms of DCA.

Table 1: **Results of ablation study.** The Top-n success rate (SR) results are compared.

| Ablation | Top-n SR |
|---|---|
| LaMPSite | **66.02** |
| w/o clustering | 65.18 |
| w/o merging $z$ & $z'$ | 63.50 |
| w/o interaction module | 62.40 |

accuracy and robustness of LaMPSite as a binding site prediction method. In terms of inference time, LaMPSite exhibits an impressive performance, requiring only approximately 0.2s per query protein. This efficiency highlights the potential for LaMPSite to be employed in virtual ligand screening applications.

We also explore how LaMPSite performs when replacing the protein contact maps with the corresponding experimental protein holo-structures. We denote the resulting model as LaMPSite[holo]. As depicted in Figure 2(b), LaMPSite[holo] achieves a success rate of 67.96%, which is comparable to DeepPocket and P2Rank. This is expected, given that protein contact maps contain only low-resolution geometric information compared to experimentally determined protein structures, and the contact maps generated from ESM-2 are derived from unsupervised training on a limited set of 20 proteins, offering only a coarse estimation of protein structures.

**Visualizations of Predicted Residues.** In Figure 3, we show examples (PDB IDs: 1O86, 1XVT, 2G25, 2GWH) illustrating our predicted binding site residues. For 1O86, LaMPSite accurately identifies the correct binding site, which manifests as hollow spaces within the protein, as depicted in Figure 3(a). In the case of 1XVT, a protein with two domains connected by linkers, LaMPSite successfully identifies the binding site within the larger domain, situated in the upper right part of Figure 3(b). Regarding 2G25 which has two binding sites for a ligand, LaMPSite can successfully detect all of them as shown in Figure 3(c). For 2GWH that binds to two different ligands on different binding sites, LaMPSite correctly predicts the binding site based on the corresponding ligand as input individually, which demonstrates the ability of LaMPSite to discriminate binding pockets on the same protein based on the ligand.

**Ablation Study.** To evaluate the effectiveness of the modules in LaMPSite, we conduct an ablation study by creating three LaMPSite variants: one without the clustering step during inference, another without the merging of interaction embeddings $z$ and $z'$, and a third without the interaction module. The results are presented in Table 1, revealing that the original LaMPSite outperforms all ablated variants. This observation demonstrates the significance of incorporating all these modules into the model for optimal performance.

**Limitations.** While our model has demonstrated good performance, it is important to acknowledge several limitations. First, our model typically generates only one binding site candidate per prediction in most cases (a total of 383 candidates for 359 pairs in COACH420), constrained by the current filtering and clustering process during inference. Second, because our model takes a single chain as

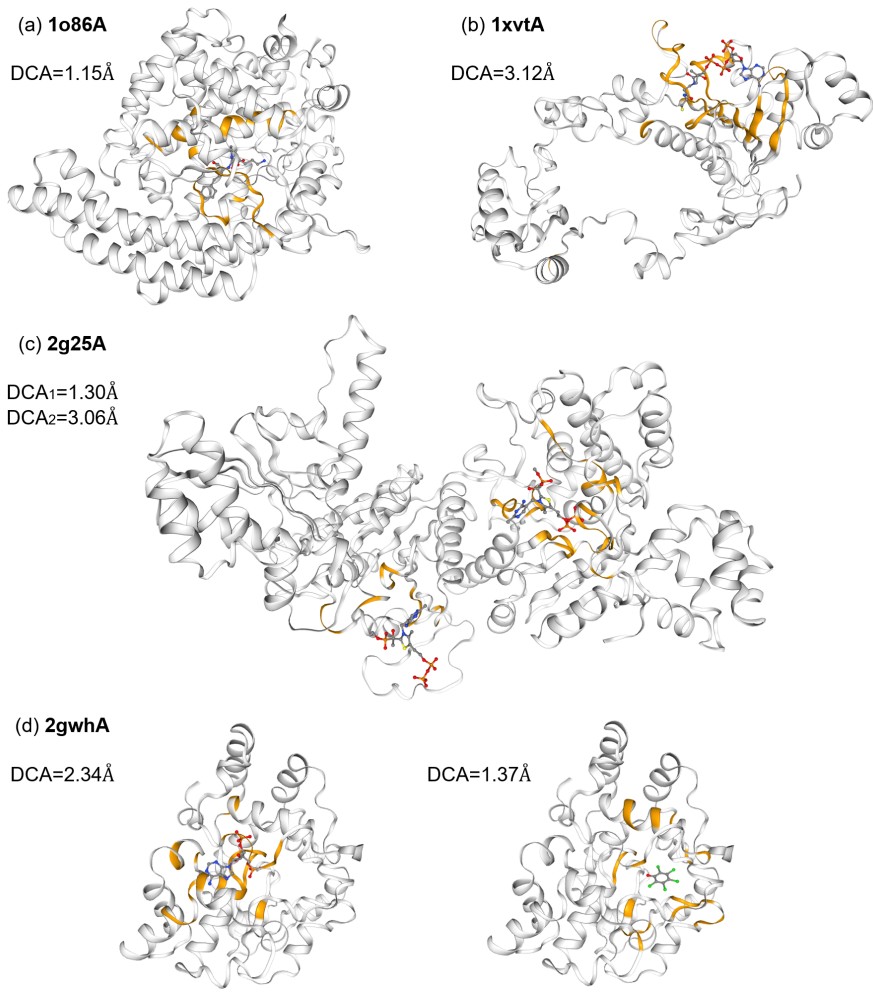

Figure 3: **Predictions and visualizations of binding sites in example complexes.** The binding site residues predicted by LaMPSite are highlighted in orange on the ground-truth 3D complex structures. The respective DCAs are also included for reference.

input, its performance in multi-chain scenarios may be limited, as it doesn't account for interactions between chains. Third, due to memory constraints, our model is not trained on relatively long protein sequences (length > 850), potentially impacting its performance. Lastly, the evaluation of our model could be expanded to encompass additional datasets and more challenging scenarios, such as the detection of cryptic binding sites in apo-structures.

## 5 Conclusion

In this work, we introduce LaMPSite, a ligand binding site prediction model empowered by a pre-trained protein language model. By relying solely on protein sequences and ligand molecular graphs as initial input, LaMPSite eliminates the need for 3D protein coordinates during the prediction process, presenting a novel approach to ligand binding site detection model design. LaMPSite demonstrates competitive performance when compared to baselines that rely on experimental 3D protein holo-structures on benchmark dataset. Future directions for research include optimizing memory utilization, assessing scenarios involving multi-chain proteins, evaluating the model's ability to detect cryptic binding sites, and integrating computationally predicted protein structures into the model.

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

# A Appendix

## A.1 Details of the Clustering and Ranking Module

In Figure A.4, we illustrate the detailed steps in the Clustering and Ranking Module of LaMPSite.

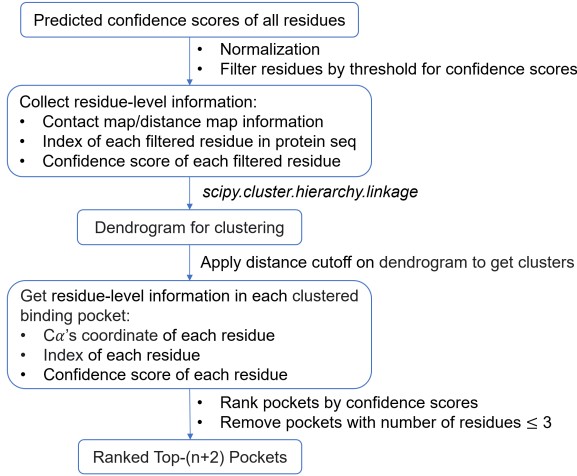

Figure A.4: Illustration of the clustering and ranking steps in LaMPSite.

## A.2 Dataset Sources

We use the publicly available data for scPDB[1] and COACH420[2]. For the data split of scPDB and the screened data entries of COACH420, we use the source[3] provided by [26].

## A.3 Implementation Details

In LaMPSite, we use the ESM-2-650M model [33] for generating protein representations. The residue embeddings from the last layer are utilized. For the Graph Neural Network [43, 38] that serves as the ligand encoder, we omit the local message passing module for efficiency. The Automatic Mixed Precision package (torch.amp) in Pytorch is used to enable mixed precision of our model for reducing memory consumption while maintaining accuracy. We use NGLview [44] to generate the visualizations of our predicted residues in Figure 3. All of the experiments are done on an NVIDIA Tesla V100 GPU (32 GB). In Table A.2, we list the typical hyperparameters used in our experiments.

Table A.2: List of hyperparameters.

| Hyperparameters | Value |
| --- | --- |
| Batch size | 8 |
| Hidden dim. in GNN | 128 |
| Hidden dim. in interaction module | 32 |
| Learning rate | 5e-4 |
| Number of GNN layers | 4 |
| Number of interaction modules | 2 |
| Max. Number of Epochs | 30 |
| Patience for early stopping | 4 |
| Dropout rate in interaction module | 0.25 |
| Threshold $v$ | 0.63 |

---

[1]http://bioinfo-pharma.u-strasbg.fr/scPDB/
[2]https://github.com/rdk/p2rank-datasets
[3]https://github.com/devalab/DeepPocket

