# OpenReview forum: "Protein Language Model-Powered 3D Ligand Binding Site Prediction from Protein Sequence"
_NeurIPS.cc/2023/Workshop/AI4Science — NeurIPS2023-AI4Science Poster_

### Official Review · Reviewer_rga8 · 2023-10-07
**Notable Efforts Toward Protein Structure-Free Ligand Binding Site Prediction**

**Rating:** 6
**Confidence:** 4

**Review:**

1. **Summary**: The authors propose LaMPSite, a new method to predict protein binding site residues in protein-ligand complexes without requiring 3D protein structural inputs, by leveraging the sequence and structural knowledge of protein language models (PLMs) such as ESM2. The results of the authors' experiments seem to suggest that:
    * (I.) ESM2's single and pairwise protein residue embeddings, along with GNN and RDKit-based ligand graph embeddings, provide competitive performance for protein-ligand binding site prediction compared to baseline methods that require 3D protein structure inputs; and
    * (II.) That providing methods such as LaMPSite with ground-truth protein distance maps improves binding site prediction performance only to a certain degree.

   Overall, these results are interesting findings for the field and should set the stage for future works to explore even more nuanced questions regarding the behavior of PLMs in different structural interaction contexts.
2. **Strengths and Weaknesses**:
   * Points of strength:
     - The authors' experiments are well-motivated, thorough, and nicely organized. It is easy to follow the authors' line of reasoning guiding the design of their experiments. It's also nice to see the ablation studies the authors included.
     - The way the authors have framed binding site prediction, in particular without requiring protein structural inputs to achieve good results, is forward-thinking and paves the way for many new approaches to tackling the binding site prediction problem.
   * Points for improvement:
     - My main concern with this work is that it seems (to me) that LaMPSite does not make full use of their RDKit conformers' initial 3D structures for each input ligand. From what I understand, the authors use a standard (invariant) graph neural network (GNN) to learn embeddings of each input ligand's molecular graph and combine those with the corresponding RDKit conformers' distance maps. To me, a more promising future direction to explore would be to use an equivariant GNN architecture to simultaneously learn not only ligand molecular graph embeddings but also equivariant (i.e., vector-valued) features directly using RDKit 3D conformer inputs. Even though the conformer structures will not be perfectly accurate 3D ligand structures, they should be useful for learning hybrid molecular graph-geometric vector feature embeddings for each input ligand.
     - The authors' details regarding clustering and ranking at the end of LaMPSite's computational pipeline are sparse and should be expanded. Otherwise, it is difficult for readers and reviewers such as myself to fully understand where the strengths and weaknesses of such clustering algorithms may lie in the context of binding site prediction.
3. **Recommendation**: Given the authors' interesting and promising efforts towards protein structure-free ligand binding site prediction, I am inclined to **weakly accept** this work.
4. **Rationale behind Recommendation**: Given the modest novelty of the authors' proposed method (i.e., combining ESM embeddings with molecular graph embeddings and RDKit conformers for binding site prediction), I currently think a score of 6 for this work is fair.
5. **Questions**:
   (1) Related to my inquiries above, how does the clustering algorithm work, and how might it affect the performance of prediction methods such as LaMPSite (and subsequently, how might it be improved)?
6. **Feedback**: I personally find the authors' efforts towards protein structure-free ligand binding site prediction encouraging and would recommend the authors continue to improve LaMPSite through strategies such as (1) including equivariant methods for learning ligand embeddings, (2) improving their clustering and ranking algorithms, and (3) incorporating ESMFold-predicted (apo) protein structures (which today are widely available and accessible for most proteins out there) into LaMPSite in a meaningful (e.g., equivariant) way.
7. **Submission Type**: The authors' manuscript successfully complies with the four to eight-page requirement for the workshop's submissions. Great work!

---

### Official Review · Reviewer_7W21 · 2023-10-19
**Review of Protein Language Model-Powered 3-Dimensional Ligand Binding Site Prediction from Protein Sequence**

**Rating:** 6
**Confidence:** 4

**Review:**

The authors present a method using ESM embeddings to predict to predict ligand-specific binding sites on a protein sequence.
Strengths (1) The authors develop a ligand-conditioned method, which is lacking in many related works. (2) The method is quite sensible, even if not technically novel. (3) The results are decent and demonstrate that PLMs have enough information to almost fully recapitulate 3D-based binding pocket predictors.
Weaknesses (1) The authors should discuss and compare with existing PLM-based binding site predictors, even if not ligand-specific. (2) The evaluations would be better if they demonstrated an ability to discriminate binding pockets on the same protein based on the ligand.

---

### Meta-Review · Area_Chair_UGB4 · 2023-10-26

**Recommendation:** Accept (Poster)
**Confidence:** 4

**Metareview:**

This paper presents a method that leverages protein language models to predict protein binding site residues in protein-ligand complexes. The reviewers are in agreement that the work is well motivated and the results are promising. They also raised a few concerns that should be taken seriously. Recommendation: Poster.